# Hydropriming and Nutripriming of Bread Wheat Seeds Improved the Flour’s Nutritional Value of the First Unprimed Offspring

**DOI:** 10.3390/plants12020240

**Published:** 2023-01-05

**Authors:** Miguel Baltazar, David Oppolzer, Ana Carvalho, Irene Gouvinhas, Luis Ferreira, Ana Barros, José Lima-Brito

**Affiliations:** 1Centre for the Research and Technology of Agro-Environmental and Biological Sciences (CITAB), University of Trás-os-Montes and Alto Douro (UTAD), Quinta de Prados, 5000-801 Vila Real, Portugal; 2Inov4Agro-Institute for Innovation, Capacity Building and Sustainability of Agri-food Production, UTAD, Quinta de Prados, 5000-801 Vila Real, Portugal; 3Plant Cytogenomics Laboratory, Department of Genetics and Biotechnology, Ed. Blocos Laboratoriais, UTAD, Quinta de Prados, 5000-801 Vila Real, Portugal; 4Department of Zootechnics, UTAD, Quinta de Prados, 5000-801 Vila Real, Portugal; 5Department of Agronomy, UTAD, Quinta de Prados, 5000-801 Vila Real, Portugal

**Keywords:** amino acids, high-performance liquid chromatography with fluorescence detection (HPLC-FLD), high-performance liquid chromatography with pulsed amperometric detection (HPLC-PAD), micronutrients, protein content, starch, sugars

## Abstract

Seed hydropriming or nutripriming has been used for wheat biofortification. Previously, the untreated S1 offspring of bread wheat S0 seeds hydro- and nutriprimed with FeSO_4_.7H_2_O and/or ZnSO_4_.7H_2_O showed improved yield relative to the offspring of untreated S0 seeds. We hypothesize that such improvement would have its origin in the higher quality of S1 seeds resulting from plants whose seeds were primed. In this work, we characterised biochemically the whole-wheat flour of unprimed S1 offspring whose S0 seeds were hydro- and nutriprimed with Fe and/or Zn and compared it to the offspring of untreated S0 seeds (control). We identified and quantified 16 free amino acids and five soluble sugars per offspring using high-performance liquid chromatography and the Association of Official Analytical Chemists (AOAC) methods. The most abundant amino acids were glutamic acid and glutamine, proline, and glycine, presenting their highest contents in the offspring of seeds nutriprimed with 8 ppm Zn (0.351 mmol∙g^−1^), 8 ppm Fe + 8 ppm Zn (0.199 mmol∙g^−1^), and (0.135 mmol∙g^−1^), respectively. The highest contents of glucose (1.91 mg∙g^−1^ sample), ash (24.90 g∙kg^−1^ dry matter, DM), and crude protein (209.70 g∙kg^−1^ DM) were presented by the offspring resulting from 4 ppm Fe + 4 ppm Zn, 8 ppm Zn, and 8 ppm Fe + 8 ppm Zn, respectively. The highest total starch content (630.10 g∙kg^−1^ DM) was detected in the offspring of seeds soaked in 8 ppm Fe. The nutritional value of the flour of the S1 offspring resulting from nutripriming was significantly higher than the control. Overall, the novelty of our research is that seed priming can improve the quality of the wheat grain and flour, at least till the first offspring, without the need to repeat the presowing treatment. Beyond the study of subsequent generations, the unravelling of transgenerational mechanisms underlying the biochemical improvement of the offspring is approached.

## 1. Introduction

Bread wheat (*Triticum aestivum* L.) is essential worldwide due to its nutritional and economic value, supplying one-third of the world’s population with more than half of their daily calorie intake [1,2]. One of the major priorities of the existing wheat-breeding programs is to improve its nutritional value by biofortification despite the challenge of combining quantity (grain yield) and flour quality [3,4,5]. Plant breeding (genetic biofortification), fertilization (agronomic biofortification), transgenesis, plant-growth-promoting soil microorganisms that enhance the mobility of micronutrients from soil to plants, and more recently, gene and genome editing, and high-throughput phenotyping, have been used to improve the nutritional status of various crops [3,4,6,7].

The imbibition of bread wheat seeds with aqueous solutions containing the micronutrients iron (Fe) and zinc (Zn) has been used to improve the germination, seedling emergence, vegetative growth, development, stress response, and nutritional quality of the grain [3,4,5,6,7,8,9,10]. These two micronutrients are essential to plants, having crucial roles in metabolic, physiological, and molecular processes [5,10,11,12,13,14]. Zinc is a cofactor of various enzymes, some of which have an antioxidant function, and is required for the production of tryptophan and other amino acids; it is a precursor of the phytohormone auxin, and is involved in protein and chlorophyll synthesis and pollen formation [5]. Iron has a pivotal role in DNA synthesis, respiration, and photosynthesis, due to its participation as a component of enzymes such as the cytochromes of the electron transport chain [13]. Iron is also involved in chlorophyll synthesis and contributes to chloroplast structure and function maintenance [13].

Seed priming with micronutrients (nutripriming) has been used to overcome micronutrient deficiency in crops [15,16,17,18,19]. Nutripriming consists of imbibing seeds with micronutrient-rich solutions before germination and sowing. This affordable method can increase the content of the micronutrients in the edible portions of the plant, the yield and/or protein amount in various crops, including wheat [9,19,20,21,22,23,24,25,26,27]. However, such benefits depend on the use of suitable micronutrient doses since their use in excess may cause cyto- and/or phytotoxicity [23,24,28,29,30], due to the osmotic stress and redox imbalance that cause changes in biochemical compounds such as total soluble sugars, amino acids, protein, phenolics, hormones, and lipids [31,32].

Our previous cytogenetic works revealed a certain degree of cytotoxicity in bread wheat seeds nutriprimed with 4 and/or 8 ppm of Fe and/or Zn [23] but also a high protein content [24]. The unprimed offspring of these plants (S1) were cytogenetically analysed and revealed less cytotoxicity than the S0 generation [25]. Moreover, the S1 offspring presented an agronomic performance that surpassed the S0 plants and the control (S1 offspring of untreated S0 seeds) [25]. Therefore, we hypothesised, for the present work, that such improvement would have its origin in the higher nutritional value of the unprimed S1 seeds that were harvested in plants whose seeds (S0) were nutriprimed with Fe and/or Zn and hydroprimed by [23], relative to the S1 offspring of untreated S0 seeds. Despite the wide availability of works related to seed priming with Fe or Zn and other micronutrients used alone or in combination, such a procedure is carried out every sowing season. We did not find in the literature any work that prevailed in the study of the effects of seed priming in the unprimed offspring. Hence, in this work, we aimed to characterise biochemically the whole-wheat-flour samples of unprimed S1 offspring that resulted from S0 plants whose seeds were hydroprimed and nutriprimed with 4 and/or 8 ppm of Fe and/or Zn, using as a control an S1 offspring of unprimed S0 seeds. Thus, we intend to infer the transmission of seed priming benefits that improve the nutritional value of the whole-wheat flour of the progeny without the need for treatment repetition before sowing.

## 2. Results

### 2.1. Identification and Quantification of Free Amino Acids

Most of the S1 offspring showed higher concentrations of each amino acid than the control (Table 1).

In all S1 offspring, the most abundant amino acids were the combination of glutamic acid and glutamine, proline, and glycine, whereas tryptophan showed the lowest average concentrations (Table 1). Apart from threonine, the average concentrations of all amino acids showed significant differences (*p* < 0.05) among the S1 offspring (Table 1). The S1 offspring of S0 seeds nutriprimed with 8 ppm Fe + 8 ppm Zn showed the highest mean values of nine out of the 16 identified amino acids, and six of them (histidine, threonine, valine, isoleucine, leucine, and phenylalanine) (Table 1) are essential to the human diet [4,33].

The S1 offspring of 4 ppm Fe + 4 ppm Zn presented higher mean values of aspartic acid + asparagine, alanine, and lysine, and those nutriprimed only with Fe showed the highest serine concentrations (Table 1). Histidine content was higher in the S1 offspring resulting from nutripriming with 8 ppm Zn, 8 ppm Fe + 8 ppm Zn, and 4 ppm Fe + 4 ppm (Table 1). The highest tryptophan concentration (0.01 mmol∙g^−1^) was found in the S1 offspring of S0 seeds primed with 4 ppm Zn (Table 1).

### 2.2. Concentration of Soluble Sugars

The average concentration of total sugars significantly differed (*p* < 0.05) among the S1 offspring (Table 2).

The highest content of total sugars was found in the control, whilst the lowest one was found in the S1 offspring of 8 ppm Fe + 8 ppm Zn (Table 2). The glucose content was higher in all S1 offspring than in the control (Table 2). Most of the S1 offspring presented a significantly (*p* < 0.05) lower concentration of sucrose, fructose, raffinose, and maltose than the control (Table 2). The S1 offspring of 8 ppm Fe + 8 ppm Zn and 8 ppm Fe presented higher average raffinose concentrations than the control (Table 2).

### 2.3. Ash, Crude Protein, and Total Starch Concentrations

The average concentration of ash, crude protein (CP), and total starch showed significant differences (*p* < 0.05) among the S1 offspring (Table 3).

For ash, the lowest and the highest average contents were observed in the control and in the S1 offspring of 8 ppm Fe + 8 ppm Zn, respectively (Table 3). The lowest average CP content was found in the control, and the highest ones were detected in the S1 offspring of 8 ppm Fe + 8 ppm Zn and 8 ppm Zn (Table 3). The highest average concentrations of total starch were found in the S1 offspring of 8 ppm Fe and control (Table 3). The lowest average content of total starch was found in the S1 offspring of 8 ppm Fe + 8 ppm Zn (Table 3).

## 3. Discussion

Previous cytogenetic, morphological, and yield-related parameter analyses were performed in bread wheat roots and plants of the S0 generation that resulted from nutripriming with 4 and/or 8 ppm of Fe and/or Zn. Upon nutripriming of the S0 seeds, in comparison to the hydroprimed ones, we verified in our previous studies that: (i) the meristematic root cells showed a higher degree of cytotoxicity, and (ii) the mature S0 plants presented lower yield-related parameters; however, (iii) their whole-wheat-flour samples had a higher amount of total soluble protein content [23,24]. In addition, the unprimed S1 offspring from nutripriming revealed lower cytotoxicity in the meristematic root cells and higher yield than the corresponding S0 plants and the control (S1 offspring of untreated S0 seeds) [25]. These previous results allowed us to hypothesise that the nutripriming and hydropriming performed in the S0 seeds increased the nutritional value of the seeds of the unprimed S1 offspring contributing to their yield improvement reported by [25]. To confirm such a hypothesis, we developed the present work, which consisted of the biochemical characterisation of whole-wheat-flour samples of the unprimed S1 offspring resulting from S0 seeds that were hydroprimed and nutriprimed with 4 and/or 8 ppm of Fe and/or Zn, in comparison with one unprimed S1 offspring of untreated S0 seeds (control).

The biochemical characterisation focused on the identification and/or quantification of free amino acids, soluble sugars, ash, crude protein (CP), and starch.

### 3.1. Amino Acids Profile and Protein Content

We identified 16 free amino acids in the whole-wheat-flour samples of the S1 offspring. Eight of them (histidine, threonine, valine, lysine, isoleucine, leucine, phenylalanine, and tryptophan) are essential amino acids (not synthesised by animals) and are required in adequate amounts to the human diet [4,33]. The S1 offspring of 8 ppm Fe + 8 ppm Zn presented high concentrations of six of these eight essential amino acids, namely, histidine, threonine, valine, isoleucine, leucine, and phenylalanine.

Beyond the improvement of the quality of whole-wheat flour, the amino acids also contribute to the proper development of the plant, given their crucial roles in various biological processes [5,13]. Amino acids are used as the building blocks of proteins and are involved in signalling methods and stress response [34,35,36,37,38].

The profile of the free amino acids present in wheat flour differs among varieties, but the high content of glutamic acid and/or glutamine is common [37,39,40]. As verified in this work, the variety “Jordão” is not an exception since a high content of glutamic acid and glutamine was detected in the unprimed S1 offspring of untreated (control), hydroprimed, and nutriprimed S0 seeds, being higher in the latter ones. Glutamic acid is synthesised from α-ketoglutarate and other amino acids such as ornithine, proline, arginine, and glutamine [37]. Therefore, the high levels of proline and glutamine probably contributed to the high amount of glutamic acid in the analysed wheat flour samples. Furthermore, proline and glutamine are the functional amino acids for dough formation [37]. The bread wheat “Jordão” has been reported as one high-quality wheat for baking [41]. Another important feature related to the baking use of wheat flour is the amount of free asparagine and its role in forming carcinogenic acrylamide during high-temperature cooking and processing [14]. The concentration of free asparagine in crops may vary with micronutrient availability, environment, genotype, and crop management or can be accumulated at high concentrations (along with proline and glycine) in response to stress, influencing the yield [14,42,43]. All S1 offspring showed an increase in the average concentration of aspartic acid and asparagine. Still, only those resulting from priming with 4 ppm Fe + 4 ppm Zn, and Zn alone, differed from the control. The combination of aspartic acid and asparagine was not among the most abundant free amino acids quantified in the samples analysed here. Asparagine and aspartic acid are also beneficial and are associated with nitrogen transport and recycling, storage, and metabolism in plants [44,45]. Other amino acids, such as serine, threonine, arginine, glutamine, histidine, glycine, isoleucine, leucine, and tryptophan, are involved in crucial pathways, proteins, and enzymes that are upregulated during stress [46,47,48,49,50]. Leucine, valine, and isoleucine increased significantly in all S1 offspring, relative to the control. These essential branched-chain amino acids (BCAA) promote energy metabolism (glucose uptake, mitochondrial biogenesis, and fatty acid oxidation), improve protein synthesis, and/or inhibit protein degradation in mammals [51]. The amino acid content highly determines the wheat grain’s nutritional value and protein content, but it is influenced by environmental conditions [37]. High temperature and drought shorten the grain filling period and influence the amino acid composition by increasing the content of phenylalanine, glutamine, and proline and decreasing other amino acids due to the accumulation of gliadins, albumins, and globulins [37,52]. The storage protein gliadins are rich in glutamine and proline and accumulate to the detriment of albumins and globulins that have structural and metabolic roles and are rich in threonine, lysine, methionine, valine, and histidine [37]. The amount of storage proteins increases in a later stage of grain development [53,54], explaining the higher amounts of glutamine and proline found in the flour samples of the S1 seeds. Proline was one of the most abundant free amino acids detected in the samples analysed here. Proline accumulates in stressed plants, enhancing their tolerance by decreasing lipid peroxidation, improving membrane, protein, and enzyme stability, and raising the protease activity with a potential decline of protein content [31,55]. Some authors reported a negative correlation between the contents of essential amino acids and proteins in stressed plants [37]. However, all the S1 offspring resulting from hydro- and nutriprimed S0 seeds showed concentrations of essential amino acids and CP contents higher than those verified in the control (S1 offspring of untreated S0 seeds).

In our previous study, the S0 seeds primed with 4 ppm Fe + 4 ppm Zn, 8 ppm Fe + 8 ppm Zn, and 8 ppm Zn, showed a higher content of total soluble protein than the hydroprimed ones used as a control in this study [24]. Similar results were found in the present work compared with the S1 offspring of hydroprimed S0 seeds. This latter offspring presented a higher CP content than the control used in the present study (offspring of untreated S0 seeds). Such a result may be due to an enhanced secretion of hydrolytic enzymes [55] that degrade storage proteins into soluble peptides and free amino acids [31] required for embryo growth [56], and that is triggered by hydropriming. Moreover, the lowest CP content was shown by the control. Therefore, hydropriming and nutripriming performed with Fe and/or Zn solutions in the S0 seeds [23,24] enhanced the CP content of the respective S1 offspring (this work). The protein amount of wheat grain varies with genotype and environment [37,57,58]. Punia et al. [59] classified wheat cultivars according to their protein content as high protein cultivars (HPC) with more than 12% CP, medium protein cultivars (MPC) with 10–12% CP, and low protein cultivars (LPC) with less than 10% CP. The control S1 offspring presented an average CP of 12.06% (120.60 g∙kg^−1^ dry matter), thus placing “Jordão” in the classification between MPC and HPC [59]. The remaining S1 offspring, resulting from hydro- and nutripriming of the S0 seeds, presented CP values higher than 12%, placing “Jordão” into the HPC category. Among the S1 offspring resulting from nutripriming, those whose S0 seeds were treated with Zn alone, and Fe + Zn, presented higher CP content, probably due to the higher mobility of Zn in phloem [60] and its pivotal role as an activator and cofactor of the metalloenzymes involved in carbohydrate metabolism and protein synthesis [6,31].

### 3.2. Soluble Sugars and Total Starch Content

The content of total soluble sugars decreased in all S1 offspring except in the control. This decrease significantly differed between the control and the following S1 offspring: hydropriming, Zn priming, and 8 ppm Fe + 8 ppm Zn. Among the five sugars identified, only glucose increased relative to the control. This increase was significant in the S1 offspring of 4 ppm Fe + 4 ppm Zn, 8 ppm Zn, and 8 ppm Fe + 8 ppm Zn. Such an increase in glucose might be related to the stress responses of the plant during its life cycle, as previously reported for other wheat cultivars [61]. Furthermore, a high amount of glucose in the mature grain is expected for glycolysis, which constitutes the first step of respiration needed for germination [60].

The average sucrose, fructose, raffinose, and maltose concentrations in the S1 offspring was lower than in the control. Seed priming improves the seedling dry weight, leaf area, and leaf CO_2_ net assimilation, maximizes the photochemical efficiency of photosystem II, and α-amylase activity under abiotic stress [62]. Therefore, the enhanced amylase activity during the maturation of primed seeds may result in higher oligosaccharide degradation, explaining the reduction of the above-mentioned sugars.

The lowest amounts of total starch, accompanied by the high content of glucose detected in the S1 offspring of S0 seeds hydroprimed and nutriprimed with Fe + Zn, reflected the enhanced activity of the enzymes responsible for starch breakdown, such as α- amylase, β-amylase, or α-glucosidase, also explaining their high CP content.

In addition to the reduced amount of total sugars, we verified low antioxidant capacity (data not shown) and increased ash and CP concentrations. Our results are in concordance with previous studies, which reported that the increase in minerals (including Fe and Zn) in wheat flour could be accompanied by a significant decrease in the antioxidant capacity and carbohydrate content and by the augmentation of protein, fibre, ash, and fat concentrations [31,63]. Furthermore, HPC wheat cultivars may have a low carbohydrate content [59].

## 4. Materials and Methods

### 4.1. Solvents and Chemicals

All chemicals and reagents were of analytical grade unless specified otherwise. The amino acid standards: L-alanine (batch no. 66H0292), L-arginine (batch no. 16H1490), L-aspartic acid (batch no. 57H11401), glycine (batch no. 36H2503), L-histidine (batch no. 27H04061), L-isoleucine (batch no. 45H0145), L-leucine (batch no. 55H0786), L-norvaline (batch no. 105H1127), L-phenylalanine (batch no. 47H1485), L-serine (batch no. 57H0508), L-threonine (batch no. 87H0098), L-tyrosine (batch no. 57H11041), L-tryptophan (batch no. 27H12431), and L-valine (batch no. 35H0531), were purchased from Sigma-Aldrich (Steinheim, Germany). The amino acid standard L-glutamic acid (batch no. K14508091) was acquired from Merck (Darmstadt, Germany), and the L-lysine (batch no. 13 ID 0627/0) was from Extrasynthese (Genay, France).

The 6-aminoquinolyl-N-hydroxysuccinimidyl carbamate (AQC, batch no. D2716) was acquired from Santa Cruz Biotechnology, Inc. (Heidelberg, Germany). Acetonitrile (UPLC grade, batch no. J113A) was purchased from Honeywell (Offenbach, Germany). Sodium hydroxide (NaOH, batch no. 0001227528) and hydrochloric acid (HCl, batch no. 001538985) were obtained from Panreac (Castelar del Vallés, Barcelona, Spain).

Calcium disodium ethylene diamine tetraacetic acid (EDTA batch no. 11466302) was acquired from Scharlau (Sentmenat, Spain).

Sodium acetate anhydrous (batch no. SZBF2210V), sodium tetraborate (Na_2_B_4_O_7_, batch no. 129H0052), triethylamine (TEA, batch no. BGBB6387V), and phosphoric acid (batch no. SZBF0350V), were purchased from Sigma-Aldrich (Steinheim, Germany).

### 4.2. Plant Material and Sample Preparation

For this study, we used untreated S1 seeds of bread wheat (T. aestivum) variety “Jordão” that constitute the offspring of S0 seeds that were kindly provided by José Coutinho (“Instituto Nacional de Investigação Agrária e Veterinária”—INIAV, I.P., Elvas, Portugal).

The S0 seeds were previously hydroprimed (soaking in distilled water), and nutriprimed with aqueous solutions of 4 ppm or 8 ppm of iron (II) sulfate heptahydrate (FeSO_4_.7H_2_O, batch no. 20K044109, VWR International, Leuven, Belgium), or zinc sulfate heptahydrate (ZnSO_4_.7H_2_O, batch no. 270313, Fluka Chemie AG, Buchs, Switzerland), as described by [23]. The untreated S1 offspring resulting from these treatments (Table 4) were used in the present work and were named throughout the text as unprimed S1 offspring.

The S1 seeds were stored for 9 months at −20 °C and then allowed to air dry for 24 h before being milled and lyophilised at −80 °C under 200 mT for 4 days. The lyophilised whole-wheat-flour samples were maintained within an exicator during the realisation of biochemical analyses.

Each whole-wheat-flour sample corresponded to the seeds harvested in four S1 plants per offspring, including the control (S1 seeds that constitute the offspring of untreated S0 seeds). Per S1 offspring, we used three whole-wheat-flour samples. Therefore, the 24 whole-wheat-flour samples (8 treatments × 3 replicates) that were biochemically analysed corresponded to S1 seeds of 12 different plants per offspring.

### 4.3. Amino Acids Determination by High-Performance Liquid Chromatography with Fluorescence Detection (HPLC-FLD)

To determine the free amino acid concentrations, except for tryptophan and tyrosine, 0.025 g of each flour sample was added to 5 mL of 6 mol∙L^−1^ HCl and tightly sealed with a cap. Hydrolysis was carried out at 110 °C for 24 h, after which samples were left to cool at room temperature (RT) and adjusted to pH 2.0 with NaOH. To each sample, 1 mL of 5 mmol∙L^−1^ of the internal standard L-norvaline solution was added and completed with distilled water till reaching a final volume of 5 mL in a volumetric flask. An aliquot of 1 mL was then filtered through a 0.22 µm filter and stored at 4 °C until initiating the analysis. For tryptophan and tyrosine, 0.025 g of each sample was added to 5 mL of 5 mol∙L^−1^ NaOH, and the hydrolysis occurred at 120 °C for 12 h. The samples were left to cool at RT, the pH was adjusted to 2.0 with HCl, and 100 µL of tramadol hydrochloride at 500 µg∙mL^−1^ was added. These solutions were completed with distilled water until a final volume of 50 mL within a volumetric flask. An aliquot of 500 µL was filtered through a 0.22 µm filter into an HPLC vial and stored at 4 °C until initiating the analysis. Processed samples and calibration curve standards were prepared according to the pre-column derivatization procedure using 6-aminoquinolyl-N-hydroxysuccinimidyl carbamate (AQC) [64], with slight modifications. In an HPLC vial insert, 5 µL of sample or standard along with 35µL of borate buffer mixture (0.2 mol∙L^−1^ of sodium borate and 5 mmol∙L^−1^ of calcium disodium EDTA, pH 8.8) and 10 µL of AQC (3 mg.ml^−1^ in acetonitrile) were added and immediately mixed for 30 s. Vials were then tightly capped and stored at 50 °C for 10 min, followed by placement in the autosampler system maintained at 10 °C. For tyrosine and tryptophan, no derivatization procedure was necessary, and processed samples were directly injected into the HPLC system.

The amino acid separation was carried out on an ACE 5 C 18 column (5 µm, 150 mm × 4.6 mm i.d., Advanced Chromatography Technologies Ltd., Aberdeen, Scotland). For all amino acids, except for tyrosine and tryptophan, a ternary gradient program was employed, with the mobile phases being 140 mmol∙L^−1^ of sodium acetate, 17 mmol∙L^−1^ of triethylamine, 1 mmol∙L^−1^ of EDTA in water, pH 4.95 (phase A), acetonitrile/distilled water (60:40, *v*/*v*) (phase B) and water (phase C). The program started at 100% phase A, increasing to 33% phase B and 7% phase C, for 40 min; followed by an increase to 40% phase B and reduction of phase C to 0%, for 8 min; and finally increasing to 100% phase B in 0.5 min, and maintained for 5.5 min. The column was re-equilibrated for 10 min between injections. Fluorescence detection occurred at excitation 250 nm and emission 395 nm. For tyrosine and tryptophan, a gradient program consisting of 50 mmol∙L^−1^ NaH_2_PO_4_ (phase A) and acetonitrile (phase B) was used. The program started at 5% phase B, increasing to 60% phase B for 8 min, and maintained for 1 min. The program then returned to the initial conditions for 0.5 min, which were kept for another 3.5 min for column re-equilibration. The injection volume was 5 µL, with the column oven set to 40 °C. Fluorescence detection was performed according to a timetable: excitation at 274 nm, emission at 304 nm; changing to excitation at 280 nm, and emission at 340 nm for 3 min; changing to excitation at 202 nm, and emission at 296 nm for the 5 min mark of the chromatographic run.

The chromatographic analyses were performed on a Thermo Scientific Dionex UltiMate 3000 Series system (Thermo Fisher Scientific, Inc., Waltham, MA, USA), composed of a RS quaternary pump, a WPS-3000RS autosampler (maintained at 4 °C), a TCC-3000RS column compartment (maintained at 35 °C), and a FLD-3400RS fluorescence detector (excitation and emission wavelength were set to 250 and 395 nm, respectively). Results were interpreted on the Chromeleon software version 7.2 (Thermo Fisher Scientific, Inc., Waltham, MA, USA). The amino acid identification was performed accordingly by comparison with authentic standards and their quantification calibration curves prepared and analysed anew for every day of analysis.

### 4.4. Determination of Soluble Sugars by High-Performance Liquid Chromatography with Pulsed Amperometric Detection (HPLC-PAD)

An amount of 0.100 g of each wheat flour sample was added to 6 mL of distilled water, and the extraction was performed at 80 °C for 30 min. Then, each solution was transferred to a volumetric flask, and the volume was completed to 50 mL with distilled water. An aliquot of 1 mL was filtered through a 0.22µm filter and stored at 4 °C until initiation of the analysis. The sugar separation was performed with 30 mM NaOH in a Dionex CarboPac PA200 (3 × 250 mm i.d.) analytical column (Thermo Fisher Scientific, Inc., Waltham, MA, USA). To prevent the formation of carbonates, 2 mmol∙L^−1^ of barium hydroxide was added to all mobile phases. After being filtered through a 0.22 µm membrane, the mobile phases were kept under a helium atmosphere (0.3–0.5 bar) during the entire time of analysis. The column oven was kept at 28 °C, and the injection volume was 5 µL. The sugars were detected by pulsed amperometric detection (PAD) using a 6041RS amperometric cell with gold working electrode through a quadrupole waveform: +200 mV (500 ms), −2000 mV (10 ms), +600 mV (10 ms) and −100 mV (10 ms). The chromatographic analyses were carried out on a Thermo Scientific Dionex UltiMate 3000 Series system (Thermo Fisher Scientific, Inc., Waltham, MA, USA), composed of an LPG-3400RS quaternary pump, a WPS-3000TRS autosampler (maintained at 4 °C), and an ECD-3000RS electrochemical detector with an incorporated column compartment. Results were interpreted using the Chromeleon software version 7.2 (Thermo Fisher Scientific, Inc., Waltham, MA, USA).

### 4.5. Chemical Analysis for Ash, Crude Protein, and Total Starch

The 24 lyophilised-wheat-flour samples were analysed using the official procedures described by the Association of Official Analytical Chemists [65] for ash (#942.05), crude protein (#954.01), and total starch (#996.11). For each of these analyses, three technical replicates of the three whole-wheat-flour samples per S1 offspring were made. Therefore, the results presented for each one of these analyses correspond to nine replicates.

### 4.6. Statistical Analyses

All results are presented as mean ± standard error (S.E.) values per S1 offspring (n = 3) indicated in Table 4. All data were analysed statistically by one-way ANOVA and Tukey’s honestly significant difference (HSD) test using the software IBM SPSS Statistics for Windows, Version 20 (IBM Corp., Armonk, NY, USA). The *p*-value significance was set for probabilities lower than 5% (*p* < 0.05).

## 5. Conclusions

The results achieved from the biochemical characterisation performed on the unprimed S1 offspring corroborated our experiment’s hypothesis in the sense that a high nutritional value was achieved in the whole-wheat-flour samples of those resulting from nutripriming with Fe and/or Zn, and hydropriming, relative to the control (unprimed S1 offspring of untreated S0 seeds). Furthermore, the results were consistent with our previous determination of total soluble protein content in the nutriprimed S0 seeds.

Based on our previous and present research, we recommend the use of reduced concentrations of these two essential micronutrients, Fe and Zn, to prevent cyto- and phytotoxicity and, as demonstrated here, to improve the nutritional value of the grain and whole-wheat flour in the plants resulting from seed priming and, at least, in their first offspring.

In conclusion, our work evidenced the effects of the transmission of seed priming from generation S0 to the S1 offspring without repeating the treatments on the S1 seeds. Future research regarding unravelling the mechanisms underlying the transgenerational transmission of the improved nutritional value of the grain and whole-wheat flour due to seed nutripriming is envisaged.

## Figures and Tables

**Table 1 plants-12-00240-t001:** Mean (±standard error, S.E.) concentration of free amino acids (mmol∙g^−1^) determined per unprimed S1 offspring resulting from S0 seeds that were untreated (control), hydroprimed, or nutriprimed with Fe and/or Zn. The mean values resulted from nine replicates. Values followed by different lowercase letters represent statistically significant differences (*p* < 0.05) among S1 offspring.

**Unprimed S1 Offspring Resulting from S0 Seeds That Were:**	**Concentration of Free Amino Acids (mmol∙g^−1^):**			
	**Aspartic Acid + Asparagine**	**Serine**	**Glutamic Acid + Glutamine**	**Histidine**
Untreated (control)	0.068 ± 0.010 a	0.086 ± 0.004 a	0.163 ± 0.003 a	0.017 ± 0.001 a
Hydroprimed	0.072 ± 0.011 a	0.089 ± 0.003 a,b	0.161 ± 0.003 a	0.017 ± 0.001 a
Nutriprimed with:				
4 ppmFe + 0 ppm Zn	0.073 ± 0.005 a,b	0.112 ± 0.006 c	0.233 ± 0.019 b	0.022 ± 0.001 b,c
8 ppmFe + 0 ppmZn	0.078 ± 0.007 a,b	0.117 ± 0.005 c	0.240 ± 0.018 b	0.018 ± 0.001 a,b
0 ppmFe + 4 ppm Zn	0.113 ± 0.014 b,c	0.099 ± 0.003 a,b,c	0.310 ± 0.007 c	0.023 ± 0.001 c,d
0 ppm Fe + 8 ppm Zn	0.139 ± 0.005 c,d	0.106 ± 0.005 a,b,c	0.351 ± 0.01 c	0.027 ± 0.001 c
4 ppm Fe + 4 ppm Zn	0.162 ± 0.012 d	0.111 ± 0.005 c	0.323 ± 0.009 c	0.024 ± 0.001 c,d,e
8 ppm Fe + 8 ppm Zn	0.106 ± 0.007 a,b,c	0.108 ± 0.005 b,c	0.314 ± 0.009 c	0.027 ± 0.001 d,c
*p*-value	<0.001	<0.001	<0.001	<0.001
**Unprimed S1 Offspring Resulting from S0 Seeds That Were:**	**Concentration of Free Amino Acids (mmol∙g^−1^):**			
	**Glycine**	**Arginine**	**Threonine**	**Alanine**
Untreated (control)	0.107 ± 0.008 a	0.033 ± 0.002 a	0.040 ± 0.004 a	0.059 ± 0.003 a
Hydroprimed	0.110 ± 0.009 a,b	0.034 ± 0.002 a	0.042 ± 0.004 a	0.060 ± 0.003 a,b
Nutriprimed with:				
4 ppm Fe + 0 ppm Zn	0.114 ± 0.005 a,b	0.036 ± 0.003 a	0.038 ± 0.002 a	0.067 ± 0.004 a,b,c
8 ppm Fe + 0 ppm Zn	0.105 ± 0.004 a	0.034 ± 0.003 a	0.037 ± 0.002 a	0.072 ± 0.004 a,b,c,d
0 ppm Fe + 4 ppm Zn	0.123 ± 0.004 a,b	0.046 ± 0.001 b	0.042 ± 0.002 a	0.068 ± 0.003 a,b,c
0 ppm Fe + 8 ppm Zn	0.126 ± 0.005 a,b	0.050 ± 0.002 b	0.041 ± 0.001 a	0.079 ± 0.003 b,c,d
4 ppm Fe + 4 ppm Zn	0.124 ± 0.004 a,b	0.048 ± 0.002 b	0.042 ± 0.002 a	0.090 ± 0.004 c
8 ppm Fe + 8 ppm Zn	0.135 ± 0.004 b	0.051 ± 0.001 b	0.047 ± 0.002 a	0.081 ± 0.006 c,d
*p*-value	<0.005	<0.001	>0.05	<0.001
**Unprimed S1 Offspring Resulting from** **S0 Seeds That Were** **:**	**Concentration of Free Amino Acids (mmol** **∙** **g^−1^):**			
	**Proline**	**Valine**	**Lysine**	**Isoleucine**
Unprimed S0 seeds (control)	0.108 ± 0.002 a	0.043 ± 0.002 a	0.029 ± 0.001 a	0.043 ± 0.002 a
Hydroprimed S0 seeds	0.108 ± 0.002 a	0.044 ± 0.002 a	0.030 ± 0.002 a	0.044 ± 0.002 a
Nutriprimed S0 seeds with:				
4 ppm Fe + 0 ppm Zn	0.142 ± 0.009 b	0.049 ± 0.003 c,b	0.035 ± 0.003 a,b	0.033 ± 0.002 a,b
8 ppm Fe + 0 ppm Zn	0.141 ± 0.01 b	0.047 ± 0.003 c,b	0.036 ± 0.002 c,b	0.032 ± 0.002 c,b
0 ppm Fe + 4 ppm Zn	0.189 ± 0.008 c	0.051 ± 0.001 a,b	0.038 ± 0.002 a,b,c	0.037 ± 0.001 a,b
0 ppm Fe + 8 ppm Zn	0.189 ± 0.004 c	0.054 ± 0.002 b	0.045 ± 0.002 c,d	0.039 ± 0.001 b
4 ppm Fe + 4 ppm Zn	0.176 ± 0.001 c	0.052 ± 0.002 a,b	0.047 ± 0.001 d	0.035 ± 0.001 a,b
8 ppm Fe + 8 ppm Zn	0.199 ± 0.002 c	0.056 ± 0.002 b	0.040 ± 0.002 b,c,d	0.039 ± 0.001 b
*p*-value	<0.001	<0.001	<0.001	<0.001
**Unprimed S1 Offspring Resulting from** **S0 Seeds That Were** **:**	**Concentration of Free Amino Acids (mmol** **∙** **g^−1^):**			
	**Leucine**	**Phenylalanine**	**Tyrosine**	**Tryptophan**
Untreated (control)	0.061 ± 0.003 a	0.037 ± 0.001 a	0.019 ± 0.00 a,b	0.008 ± 0.00 a
Hydroprimed	0.062 ± 0.003 a	0.038 ± 0.002 a	0.032 ± 0.001 b,c	0.009 ± 0.00 c
Nutriprimed with:				
4 ppm Fe + 0 ppm Zn	0.049 ± 0.003 c,b	0.035 ± 0.003 a,b	0.033 ± 0.002 a,b	0.071 ± 0.005 a,b
8 ppm Fe + 0 ppm Zn	0.047 ± 0.003 c,b	0.036 ± 0.002 c,b	0.032 ± 0.002 c,b	0.070 ± 0.005 c,b
0 ppm Fe + 4 ppm Zn	0.051 ± 0.001 a,b	0.038 ± 0.002 a,b,c	0.037 ± 0.001 a,b	0.085 ± 0.003 b,c
0 ppm Fe + 8 ppm Zn	0.054 ± 0.002 b	0.045 ± 0.002 c,d	0.039 ± 0.001 b	0.086 ± 0.002 c
4 ppm Fe + 4 ppm Zn	0.052 ± 0.002 a,b	0.047 ± 0.001 d	0.035 ± 0.001 a,b	0.079 ± 0.002 b,c
8 ppm Fe + 8 ppm Zn	0.056 ± 0.002 b	0.040 ± 0.002 b,c,d	0.039 ± 0.001 b	0.087 ± 0.001 c
*p*-value	<0.001	<0.001	<0.001	<0.001

**Table 2 plants-12-00240-t002:** Mean (±S.E.) concentration of soluble sugars (mg∙g^−1^ sample) determined per unprimed S1 offspring resulting from S0 seeds that were untreated (control), hydroprimed, or nutriprimed with Fe and/or Zn. The mean values resulted from nine replicates. Values followed by different lowercase letters represent statistically significant differences (*p* < 0.05) among S1 offspring.

**Unprimed S1 Offspring Resulting from S0 Seeds** **That Were:**	**Concentration of the Soluble Sugars (mg∙g^−1^ Sample):**		
	**Glucose**	**Sucrose**	**Fructose**
Untreated (control)	1.25 ± 0.11 a	13.99 ± 0.34 c	0.90 ± 0.11 b
Hydroprimed	1.62 ± 0.09 a,b	9.77 ± 0.3 a	0.70 ± 0.04 a,b
Nutriprimed with:			
4 ppm Fe + 0 ppm Zn	1.49 ± 0.06 a,b	11.93 ± 0.66 a,b,c	0.71 ± 0.06 a,b
8 ppm Fe + 0 ppm Zn	1.57 ± 0.09 a,b	10.37 ± 0.18 a,b	0.90 ± 0.05 b
0 ppm Fe + 4 ppm Zn	1.43 ± 0.08 a,b	10.96 ± 0.63 a,b	0.54 ± 0.03 a
0 ppm Fe + 8 ppm Zn	1.72 ± 0.10 a,b	11.02 ± 0.67 a,b	0.53 ± 0.04 a
4 ppm Fe + 4 ppm Zn	1.91 ± 0.27 b	11.11 ± 0.52 a,b	0.77 ± 0.15 a,b
8 ppm Fe + 8 ppm Zn	1.66 ± 0.07 a,b	12.18 ± 0.41 b,c	0.69 ± 0.02 a,b
*p*-value	<0.05	<0.001	<0.05
**Unprimed S1 Offspring Resulting from S0 Seeds** **That Were:**	**Concentration of the Soluble Sugars (mg∙g^−1^ Sample):**		
	**Raffinose**	**Maltose**	**Total sugars**
Untreated (control)	3.00 ± 0.08 b,c,d	83.34 ± 2.02 c	102.47 ± 2.18 c
Hydroprimed	2.22 ± 0.11 a	62.2 ± 1.36 a,b,c	76.5 ± 1.62 a,b
Nutriprimed with:			
4 ppm Fe + 0 ppm Zn	2.66 ± 0.1 a,b,c	76.67 ± 9.32 b,c	93.45 ± 9.78 b,c
8 ppm Fe + 0 ppm Zn	3.15 ± 0.09 c,d	67.86 ± 2.41 a,b,c	83.85 ± 2.57 a,b,c
0 ppm Fe + 4 ppm Zn	2.47 ± 0.09 a,b	58.37 ± 3.86 a,b	73.76 ± 4.24 a,b
0 ppm Fe + 8 ppm Zn	2.61 ± 0.25 a,b,c	54.02 ± 3.86 a	69.89 ± 4.01 a
4 ppm Fe + 4 ppm Zn	2.63 ± 0.22 a,b,c	68.86 ± 7.46 a,b,c	85.26 ± 7.3 a,b,c
8 ppm Fe + 8 ppm Zn	3.35 ± 0.15 d	53.3 ± 3.77 a	71.18 ± 3.91 a,b
*p*-value	<0.001	<0.05	<0.001

**Table 3 plants-12-00240-t003:** Mean (±S.E.) concentration of ash, crude protein, and total starch (g∙kg^−1^ dry matter, DM) determined per unprimed S1 offspring resulting from S0 seeds that were untreated (control), hydroprimed or nutriprimed with Fe and/or Zn. The mean values resulted from nine replicates. Values followed by different lowercase letters represent statistically significant differences (*p* < 0.05) among S1 offspring.

Unprimed S1 Offspring Resulting from S0 Seeds That Were:	Concentration (g∙kg^−1^ DM) of:		
	Ash	Crude Protein	Total Starch
Untreated (control)	18.00 ± 0.77 a	120.60 ± 0.70 a	628.40 ± 5.50 d
Hydroprimed	22.90 ± 0.65 b	188.80 ± 11.80 b,c,d	546.60 ± 6.20 a,b
Nutriprimed with:			
4 ppm Fe + 0 ppm Zn	23.10 ± 1.10 b	145.50 ± 10.30 a,b	594.60 ± 13.30 c,d
8 ppm Fe + 0 ppm Zn	22.60 ± 1.45 b	157.70 ± 13.10 a,b,c	630.10 ± 15.90 d
0 ppm Fe + 4 ppm Zn	24.40 ± 0.42 b	201.40 ± 5.40 c,d	560.90 ± 7.90 b,c
0 ppm Fe + 8 ppm Zn	24.90 ± 0.24 b	206.70 ± 8.80 d	537.90 ± 13.70 a,b
4 ppm Fe + 4 ppm Zn	22.50 ± 0.36 b	187.90 ± 10.60 b,c,d	537.40 ± 8.00 a,b
8 ppm Fe + 8 ppm Zn	25.20 ± 0.23 b	209.70 ± 11.20 d	512.80 ± 7.80 a
*p*-value	<0.001	<0.001	<0.001

**Table 4 plants-12-00240-t004:** Plant material used in this work: air-dried and unprimed S1 seeds that constitute the offspring of S0 seeds that were untreated (used as a control) and those that were hydroprimed and nutriprimed with Fe and/or Zn by [23].

Unprimed S1 Offspring Resulting from:	
Unprimed (untreated) S0 seeds	Control
Hydropriming (soaking in distilled water)	0 ppm Fe + 0 ppm Zn
Nutripriming with Fe and/or Zn (soaking in aqueous solutions of FeSO_4_.7H_2_O and/or ZnSO_4_.7H_2_O)	4 ppm Fe + 0 ppm Zn
	8 ppm Fe + 0 ppm Zn
	0 ppm Fe + 4 ppm Zn
	0 ppm Fe + 8 ppm Zn
	4 ppm Fe + 4 ppm Zn
	8 ppm Fe + 8 ppm Zn

## Data Availability

Not applicable.

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
