# Peer review of "Hydropriming and Nutripriming of Bread Wheat Seeds Improved the Flour’s Nutritional Value of the First Unprimed Offspring"

_plants, 2023, doi:10.3390/plants12020240_

Round 1
Reviewer 1 Report
The current manuscript focuses on Biochemical characterization of whole wheat flour. Authors characterized biochemically the whole wheat flour of unprimed S1 21 offspring whose S0 seeds were hydroprimed and nutriprimed with Iron sulphate (Fe) and/or Zinc 22 sulphate (Zn). Finally they concluded that concluded that the S1 off-29 spring resulting from S0 seeds nutriprimed with Fe and/or Zn showed a higher grain nutritional 30 value than the control seeds (resulting from unprimed S0 seeds). In my opinion, it is a nice piece of work. The present article was organized logically, and the objectives are apparent. Thus, the article could be considered from my point of view after revision. However, after evaluation of manuscript, the following remarks should be addressed.
1. Line-32-Amino acid can be written as “amino acid”
2. Please adjust the lettering in the table, especially in table 1. Some lettering is separated from the values and placed in the next line.
3. Please check the all concentration unit.
4. It is necessary to check grammatical and stylistic errors throughout the text. Also, it is necessary to check once again the conformity of the design of the list of references with the requirements of the journal.
Author Response
Point 1.1: Line-32-Amino acid can be written as “amino acid”
Response 1.1: The change was performed.
Point 1.2: Please adjust the lettering in the table, especially in table 1. Some lettering is separated from the values and placed in the next line.
Response 1.2: We improved the format of all tables.
Point 1.3: Please check the all concentration unit.
Response 1.3: We checked the concentration unit, consulted the MDPI instructions to authors, and published MDPI articles. In our previous version of the manuscript, we referred to the micronutrient’s unit concentration as mg.L-1 (to match our previously published articles). Accordingly, the authors’ instructions, the proper concentration unit should be presented as g∙L-1. However, after consulting published MDPI articles, we verified that ppm could also be used as a concentration unit. Therefore, we replaced mg.L-1 with ppm. We also corrected other concentration units throughout the manuscript.
Point 1.4: It is necessary to check grammatical and stylistic errors throughout the text. Also, it is necessary to check once again the conformity of the design of the list of references with the requirements of the journal.
Response 1.4: We revised the English language, grammatical and stylistic errors throughout the manuscript. We corrected the list of references accordingly in the Instructions for Authors.
Reviewer 2 Report
Dear Authors
Please find my comments in the attached file.
Regards

Author Response
Point 1. Please make title simple showing the novelty of the study
Response 1: We changed the title in the revised version of the manuscript.
Point 2. Abstract: Please follow the systematic abstract as
1- Give problem statement
2- Why current method is selected to mitigate that issue.
3- Novelty statement showing new work done in current study
Response 2: The abstract was rewritten, considering the recommendations.
Point 3. Abstract: It would be better if quantitative data is provided. Please provide that.
Response 3: Quantitative data have been added to the abstract.
Point 4. Abstract: No recommendation and future prospective is provided. Please provide these 2 important aspects.
Response 4: The revised abstract includes these aspects.
Point 5. Keywords: Please modified the keyword by referring the analysis done on HPLC.
Response 5: We replaced the keyword ‘HPLC’ with two additional keywords specifying the two HPLC analyses done in this work, namely, ‘HPLC with fluorescence detection (HPLC-FLD)’ and ‘HPLC with pulsed amperometric detection (HPLC-PAD)’.
Point 6. Line 44: Its a general statement. Be specific while writing. Please mentioned which micronutrients.
Response 6: We focused on introducing the analysed micronutrients, Fe and Zn.
Point 7. Line 46: Please define which processes
Response 7: We specified the physiological, metabolic, biochemical and molecular processes where Zn and Fe are involved.
Point 8. Lines 60-64: Please provide
1- Novelty aspect
2- Knowledge gap covered in context of literature
3-Hypothesis of experiment
Response 8: These points were included in the Introduction section.
Point 9. Results – Line 66: Its not possible to check the results and discussion parts without hypothesis. As a reviewer we have to decide the aims and null hypothesis for evaluation of results data. Please provide hypothesis first.
Response 9: The experiment’s hypothesis was included in the abstract and introduction. When we developed this work, we hypothesised that the whole wheat flour of the S1 offspring resulting from plants whose S0 seeds were nutriprimed with Fe and/or Zn would have a higher nutritional value than those resulting from hydropriming and the control. Our results, supported by statistical analyses, corroborated our null hypothesis.
Point 10. Line 240: What was company and lot number. Please mentioned any data for tractability
Response 10: We included the company number in the reagents where it was missing. We provided the batch number and other details for all the reagents used.
Point 11. Line 252: Are you sure. It can change the osmotic potential. Can also cause damage to stored food in seed. Please check and write carefully in context of osmotic-potential which seeds required and what distilled water have.
Response 11: The hydropriming treatment performed with distilled water was done in 2017, as described by Reis et al. (2018). Independently of the seed priming treatment performed in the S0 seeds, the dry seeds of all S1 offspring (S1 seeds) were stored for 9 months at -20 ºC. The S1 seeds were allowed to unfreeze at room temperature for 24h, then weighed and milled. The whole wheat flour samples were weighed before and after lyophilisation. The moisture content of the whole wheat flour samples of all S1 offspring that were eliminated by lyophilisation ranged from 3.84% to 7.13%. The lyophilized whole wheat flour samples were immediately stored within an exicator until the end of the biochemical analyses. We are aware that the moisture content affects the wheat food grain and flour. Therefore, we stored the S1 seeds at a low temperature to ensure their conservation for long-term storage. Also, independently of the seed priming treatment performed in the S0 generation, the seeds of all S1 offspring were stored under the same conditions and processed equally. Some of these details were included in the revised manuscript.
Point 12. Line 253: Which salt was used for their manufacturing. Please provide details.
Response 12: Details were provided.
Point 13. Table 4: Why these level were selected. Is there any screen data for it. If yes then provide in supplementary file. If adopted from any other published work then give reference.
Response 13: As published before, we tested 20 seed priming treatments performed with Fe and/or Zn (Reis et al. 2018). The concentrations of Fe and Zn used in the S0 seeds were not based on a specific study or criterion. We defined a random range of concentrations lower than the ones found in many articles published by various authors. In the following studies, including the present one, we focused on the highest dosages used by Reis et al. (2018), 4 ppm and 8 ppm of Fe and Zn.
Point 14. Conclusions: Please provide some recommendation and future propectives.
Response 14: Done.
Round 2
Reviewer 2 Report
Dear Authors
I am satisfied with the corrections made in the manuscript. Paper can be accepted.